# Interfacing Digital Microfluidics with Ambient Mass Spectrometry Using SU-8 as Dielectric Layer

**DOI:** 10.3390/mi9120649

**Published:** 2018-12-08

**Authors:** Gowtham Sathyanarayanan, Markus Haapala, Tiina Sikanen

**Affiliations:** Drug Research Program, Faculty of Pharmacy, Division of Pharmaceutical Chemistry and Technology, University of Helsinki, Viikinkaari 5E, 00790 Helsinki, Finland; gowtham.sathyanarayanan@helsinki.fi (G.S.); markus.haapala@helsinki.fi (M.H.)

**Keywords:** digital microfluidics, SU-8, desorption/ionization, ambient mass spectrometry, liquid-liquid extraction, drug distribution, drug metabolism, cytochrome P450, microreactor, desorption atmospheric pressure photoionization

## Abstract

This work describes the interfacing of electrowetting-on-dielectric based digital microfluidic (DMF) sample preparation devices with ambient mass spectrometry (MS) via desorption atmospheric pressure photoionization (DAPPI). The DMF droplet manipulation technique was adopted to facilitate drug distribution and metabolism assays in droplet scale, while ambient mass spectrometry (MS) was exploited for the analysis of dried samples directly on the surface of the DMF device. Although ambient MS is well-established for bio- and forensic analyses directly on surfaces, its interfacing with DMF is scarce and requires careful optimization of the surface-sensitive processes, such as sample precipitation and the subsequent desorption/ionization. These technical challenges were addressed and resolved in this study by making use of the high mechanical, thermal, and chemical stability of SU-8. In our assay design, SU-8 served as the dielectric layer for DMF as well as the substrate material for DAPPI-MS. The feasibility of SU-8 based DMF devices for DAPPI-MS was demonstrated in the analysis of selected pharmaceuticals following on-chip liquid-liquid extraction or an enzymatic dealkylation reaction. The lower limits of detection were in the range of 1–10 pmol per droplet (0.25–1.0 µg/mL) for all pharmaceuticals tested.

## 1. Introduction

Digital microfluidics (DMF), based on electrowetting-on-dielectric, is an established technology for performing automated chemical and biochemical assays in droplet scale [1]. In DMF, droplets of reagents (in the range of nL–µL) are manipulated by applying a series of voltages to an array of electrodes coated with an insulating (dielectric) layer, and further with an additional hydrophobic (e.g., a fluoropolymer) layer to reduce droplet sticking [2]. Application of voltage to individual electrodes results in charge accumulation on the insulator, thus allowing precise control of droplet movement (incl. merging, mixing, and splitting) based on changes in the surface’s electrowetting properties [3]. A variety of materials have been introduced as the dielectric layer in DMF [4], of which parylene C is by far the most widely used [5]. Although chemical vapor deposition (CVD) of parylene C results in a homogenous and defect-free layer, CVD is time-consuming and requires relatively expensive instrumentation [6]. Therefore, spin-coatable and UV curable photoresists, such as the epoxy polymer SU-8, appear as appealing alternatives to rapid and low-cost DMF microdevice fabrication, facilitating chip manufacturing even under standard laboratory conditions [7]. Owing to its good mechanical rigidity and wide thickness range, SU-8 has been the material of choice for a variety of laboratory-on-a-chip applications requiring high aspect ratio structures or ultimately sharp features, such as on-chip electrospray emitters for mass spectrometric (MS) detection [8,9,10]. With a view to DMF applications, the strong adhesion of SU-8 [11] facilitates fabrication of high quality dielectric layers by spin-coating. On the other hand, the high internal film stresses characteristic to SU-8 [11], do not play a major role in DMF, as the thickness of the dielectric layer is typically in the range of few micrometers only. Although the relatively high autofluorescence of SU-8 [9] may not favor its use in optically detected DMF assays, the possibility of lithographic patterning of the dielectric layer provides a convenient approach to incorporate, e.g., integrated metal elements (underneath the dielectric layer) into the active assay design. For example, to facilitate voltammetric analysis of the sample droplet, on-chip electrodes patterned below the dielectric have been exposed via lithographic patterning of SU-8 [12]. The good thermal stability of SU-8 has also allowed its use as the dielectric layer for high temperature polymerase chain reaction on a thermal DMF device [13]. In addition to its favorable mechanical and thermal properties, the good chemical stability of SU-8 [14] favor its use over other dielectric materials under harsh experimental conditions, e.g., when the dielectric needs to tolerate high (>100 °C) temperatures in the presence of organic solvents. An experimentally challenged approach like this is, for example, ambient mass spectrometry (MS), which would facilitate direct analysis of sample components from the surfaces of the DMF bottom plate.

Ambient MS is an analytical concept increasingly used for mapping the spatial distribution of metabolic biomarkers directly from the surfaces of biological substrates, such as human or animal tissue [15] or plant leaves [16]. The technology also allows for forensic and drug analysis directly, without additional sample preparation, from bodily fluids, fabrics, tablets, and fruit, for example [17,18]. A large variety of desorption/ionization methods have been developed to facilitate ambient MS, of which desorption electrospray ionization (DESI) and direct analysis in real time (DART) are probably the most widely used techniques in biochemical and forensic analyses, respectively. By facilitating non-targeted analysis of the assay components and reaction products in situ, ambient MS also appears as a logical, supplementary tool for monitoring of DMF-based assays in situ. However, the combination has not seen widespread exploitation among the variety of optically and electrochemically detected DMF assays, likely because it requires controlled evaporation of the sample solution prior to detection. Instead, interfacing of DMF with MS has mainly been realized via electrospray ionization (ESI) using integrated off-chip or on-chip emitters [19,20] or by transferring the sample solution to off-chip liquid chromatography (LC) MS analysis [21]. A few prior reports also exist on matrix-assisted laser desorption ionization (MALDI) MS from the DMF bottom plate [22,23]. All of the above-mentioned concepts, however, have their own limitations. For example, integration of an ESI emitter with the DMF device is technically demanding, same as the transfer of the very small volume droplet for analysis on a macro LC-MS instrument. MALDI-MS on the other hand requires substantial amount of manual sample preparation prior to analysis and is mostly feasible for large biomolecules only.

In this work, we examined the critical steps related to interfacing of DMF based biochemical assays, more precisely drug distribution (liquid-liquid extraction) and metabolism assays, with ambient MS. First of all, we chose to use desorption atmospheric photoionization (DAPPI) [24] for desorption/ionization on DMF devices. Even if DESI is the most used technique for ambient MS based bioanalysis in general, it suffers from pronounced signal suppression (loss of sensitivity) in the presence of high salt concentrations, characteristic to biochemical assays typically conducted by DMF. In terms of drug analysis, DAPPI not only provides better salt tolerance, but also enables more efficient ionization of non-polar compounds (such as most of the drugs) over DESI [25,26]. Here, we show that by exposing SU-8 spots to the assay design, to allow for controlled sample precipitation on a hydrophilic surface, the efficiency of ambient MS analysis is significantly improved over that performed on the hydrophobic (fluoropolymer) layer. Moreover, the high chemical, thermal, and mechanical stability of SU-8 are in a key role in enabling desorption/ionization directly from the SU-8 surface.

## 2. Materials and Methods

### 2.1. Materials and Reagents

Pentaerythritol tetrakis (3-mercaptopropionate), >95% (tetra-thiol), 1,3,5-Triallyl-1,3,5-triazine-2,4,6(1H,3H,5H)-trione (tri-ene), 98%, β-nicotinamide adenine dinucleotide 2′-phosphate reduced tetrasodium salt hydrate (NADPH), biotin-PEG4-alkyne, rhodamine B, magnesium chloride, paracetamol, naproxen, carbamazepine, phenacetin, testosterone, octanol, ethylene glycol, chlorobenzene, 28–30% ammonium hydroxide, 2-Amino-2-(hydroxymethyl)-1,3-propanediol (Tris), phosohate buffer saline (PBS) and chloroform were purchased from Sigma-Aldrich (Steinheim, Germany). Potassium dihydrogen phosphate and trifluoroacetic acid (TFA) were from Riedel-de Haen (Seelze, Germany). Dipotassium hydrogen phosphate was from Amresco (Solon, OH, USA). Methanol was from Honeywell Specialty Chemicals GmbH (Seelze, Germany) and Irgacure^®^ TPO-L (photoinitiator) was kindly donated by BASF (Ludwigshafen, Germany). Alexa Fluor^®^ 488 streptavidin conjugate (Alexa-SA) was purchased from Thermo Fisher Scientific (Waltham, MA, USA). The lipids 1,2-dioleoyl-sn-glycero-3-phophoethanolamine, 1,2-dioleoyl-3-trimethylammonium-propane (chloride salt), 1,2-dioleoyl-sn-glycero-3-phophoethanolamine-*N*-(Cap biotinyl) (sodium salt), and 1,2-dioleoyl-sn-glycero-3-phosphoethanolamine-*N*-(lissamine rhodamine B sulfonyl) (ammonium salt) were purchased from Avanti Polar Lipids, Inc. (Alabaster, AL, USA). Recombinant cytochrome P450 1A2 (rCYP1A2) (1000 pmol·mL^−1^) was purchased from Corning (Wiesbaden, Germany). Water was purified with a Milli-Q water purification system, (Merck Millipore, Molsheim, France). All solvents were of analytical or HPLC grade.

SU-8 5 was purchased from Microchem Corp., (Westborough, MA, USA). Fluoropolymer Cytonix FluoroPel PFC 1604V and PFC 110 fluoro-solvent were purchased from Cytonix LLC (Beltsville, MD, USA). AZ 351B resist developer was purchased from MicroChemicals GmbH (Ulm, Germany). Ormocomp was purchased from Microresist technology (Berlin, Germany).

### 2.2. Digital Microfluidic Chip Fabrication

The top- and bottom plates of DMF devices were fabricated and assembled as described previously [7]. Briefly, an array of 15 × 6 driving electrodes and 8 reservoir electrodes for sample loading were lithographically patterned on a Cr coated glass plate (Telic, Valencia, CA, USA) which served as the bottom plate of the DMF device (Figure 1a). The masked UV exposure was performed on a Dymax 5000-EC Series UV flood lamp (Dymax corporation, Torrington, CT, USA; nominal intensity 225 mW/cm^2^) followed by resist development and Cr etching under non-clean room conditions. After electrode patterning, a 8-µm-thick SU-8 layer was spin-coated (1600 rpm, 30 s) onto the electrodes, soft baked according to the standard protocol (2 min at 65 °C and 2 min at 95 °C) and flood exposed for 15 s using the Dymax lamp. After post-exposure bake (2 min at 65 °C and 2 min at 95 °C), a 1-µm-thick fluoropolymer layer was spin-coated (1% solution, 1000 rpm, 30 s) on SU-8 and baked at 150 °C for 30 min. Hydrophilic spots were made by covering the bare SU-8 layer with circular pieces of dicing tape, cut with a biopsy puncher (I.D. 1 mm), prior to fluoropolymer coating. After fluoropolymer coating, the tape was removed and the SU-8 spots were washed with methanol-water (1:1) three times and dried under nitrogen stream before use. Commercial indium tin oxide coated glass plates (Structure Probe, Inc., West Chester, PA, USA), spin-coated with fluoropolymer in the same way, were used as the top plates of the DMF devices. Droplet actuation on the DMF devices was conducted with the help of an open-source DropBot automation system [27].

For heating of the DMF based drug metabolism assays to physiological temperature (37 °C), ink-jet printed thin-film silver microheaters were used. The microheaters (R = 30.8 ± 0.4 Ω, n = 5) were printed with a Brother MFC-J5910DW printer out of a commercial silver ink (AGIC-AN01 Silver Nano Ink, AgIC Inc., Tokyo, Japan) attached to the DMF bottom plate and resistively heated as described earlier [7].

### 2.3. Mass Spectrometry

For DAPPI-MS analysis, the DMF devices were connected to a Bruker micrOTOF™ mass spectrometer (Bruker Daltonics, Bremen, Germany). The DAPPI-MS setup includes a microfabricated heated nebulizer chip for vaporizing the organic solvent jet needed for desorption, and a krypton filled vacuum UV lamp (PKR 106; Heraeus Noblelight GmbH, Hanau, Germany) for photoionization (Figure 1b). Schematic views of the DAPPI-MS setup and the heated nebulizer chip are given in the Appendix A. The heated nebulizer chip fabrication as well as optimization of the key operation parameters related to DAPPI-MS analysis are described elsewhere [24,26]. Briefly, in DAPPI-MS, the desorption/ionization process begins with thermal desorption (vaporization) of solid sample into the gas phase with help of a heated jet of vaporized organic solvent (Figure 1c). The gas phase solvent molecules are ionized with the help of photons (10 eV), after which they induce ionization of the sample components via gas phase charge exchange or proton transfer reactions. As the result, the sample components are detected on the MS as radical cations (M^+•^) or protonated ions [M + H]^+^, respectively, depending on their proton affinity and ionization energy. In this work, sample desorption/ionization from the DMF bottom plate was performed using chlorobenzene (250 °C, 10 µL/min) as the organic solvent, vaporized with help of the nebulizer chip using heating power of 4 W. In addition, nitrogen (180 µL/min) was used as an assisting (nebulizer) gas. The nebulizer chip was fixed at an angle of 45° with respect to the sample (DMF) plate (Appendix A). On the MS, the drying gas temperature was set to 300 °C and the flow rate to 4 L/min.

After performing the desired sample manipulations on the DMF device, the droplets were transferred onto the hydrophilic spots by DMF actuation. The top plate was then removed and the droplets were let dry prior to DAPPI-MS analysis (Figure 1c). The sample spots were then aligned with respect to the heated solvent jet, the UV lamp, and the MS inlet with help of an xyz-positioning stage (Figure 1b,c). The distance between the sample spot and the MS inlet was ca. 2 mm and the UV lamp was fixed ca. 10 mm above the sample spot. The DAPPI-MS method qualification (determination of the lower limits of detection) was conducted with help of three pharmaceuticals, including paracetamol, naproxen, and carbamazepine, using testosterone as an internal qualifier (Table 1). For optimization of the MS parameters, dummy glass plates without the electrodes and patterned with SU-8 (hydrophilic spots) and fluoropolymer in the same manner, were used as substrates onto which the sample droplets (1 µL) were manually dispensed. The MS data was analyzed with Compass DataAnalysis 4.0 software (Bruker Daltonics, Bremen, Germany). For data analysis (peak area integration), extraction window of ±0.05 *m*/*z* was used.

### 2.4. On-Chip Drug Distribution Assays

The on-chip drug distribution assays were based on three-phase liquid-liquid extraction (LLE) protocol as described in Figure 2a. For on-chip LLE, the sample solution (containing three pharmaceuticals and testosterone as an internal qualifier, each 45 µg/mL) was acidified with 0.1% TFA (pH 2), after which the aqueous sample droplet (donor) was encapsulated with octanol (acceptor). The first extraction (water → octanol distribution) was carried for 30 min to allow the extraction between the donor (volume equal to two electrodes) and the acceptor (volume equal to six electrodes). The process (step 2 in Figure 2a) was then repeated three times by bringing a fresh droplet of sample solution (donor) into contact with the same octanol droplet (acceptor). The second extraction (octanol → water) was carried out by bringing a droplet of ammonium hydroxide (pH 12) (acceptor, 1 electrode volume) inside the octanol (donor). The extraction time was again 30 min, after which the aqueous droplet was separated from octanol, brought onto the hydrophilic SU-8 spot and allowed to evaporate prior to DAPPI-MS analysis. During both extractions, the octanol phase was stationary and only the aqueous droplets were brought in and out of the octanol droplet. This was facilitated by the fact that octanol was immobile at frequencies used for actuation of the aqueous phase. The driving potential used was 100–140 V_rms_ at 10 kHz and 300 Hz for the aqueous phase and octanol, respectively.

### 2.5. Preparation of Immobilized Cytochrome P450 Reactors

For conducting drug metabolism reactions on the DMF devices, porous polymer monoliths functionalized with immobilized recombinant cytochrome P450 (rCYP) enzymes were prepared similar to [7]. Briefly, the porous polymer monoliths were prepared out of thiol-ene polymer and functionalized sequentially with biotin, fluorescent labeled streptavidin (Alexa-SA, ex/em 495/519 nm), biotinylated fusiogenic liposomes (b-FL), and rCYP. To prepare the porous polymer monoliths, a 0.5 g mixture of tetra-thiol and tri-ene monomers (50 mol% excess thiol) was stirred along with 5 µL of 10% Irgacure TPO-L in methanol and 2 g of methanol (porogen). While stirring, 2 µL of the emulsion was on a hydrophilic spot (2 mm dia.) patterned onto the bottom plate and cured under UV (1.5 min flood exposure) with the top-plate attached. After curing, the monolith plugs were thoroughly washed with a few mL of MilliQ water. The monoliths were then functionalized with biotin following the previously published protocol [29]. Briefly, 2 µL of 1 mM biotin-PEG_4_-alkyne in polyethylene glycol containing 1% (*v*/*v*) Irgacure TPO-L was dispensed onto the monolith and allowed to crosslink with the surface thiols under UV for 1 min, after which the monolith was sequentially washed with a few mL of methanol and phosphate-buffered saline (PBS). Next, 2 µL of 0.5 mg/mL Alexa-SA in PBS was pipetted on the monoliths and incubated for 15 min at room temperature. After washing with few mL of PBS, 2.5 µL of b-FLs was added on to the monoliths, incubated at room temperature for 30 min, followed by PBS wash. Finally, 2.5 µL of the recombinant CYP1A2 isoform stock solution was pipetted on the monoliths and allowed to fuse with the b-FLs at 37 °C for 30 min. The b-FLs were prepared and fused with rCYP similar to the previous work [30]. After the fusion, the monoliths were washed with PBS and kept wet until use.

### 2.6. On-Chip Drug Metabolism Assays

The on-chip drug metabolism reactions were performed on the monolith-based rCYP1A2 reactors using phenacetin O-deethylation as the model reaction (Figure 2b). Before initiating the CYP reaction, the storage buffer (PBS) was replaced by DMF actuation with the reaction solution containing 100 µM phenacetin as the model substrate and 0.5 mM NADPH as the co-substrate in 15 mM Tris buffer (pH 7.5) containing 0.5 mM MgCl_2_. The reaction was initiated by applying heating power of 0.1 W via the printed microheater to reach the physiological temperature (37 °C). The initial reaction volume was ca. 3 µL (equal to area of four DMF electrodes) and the evaporation due to heating was compensated by adding 1.5 µL of fresh buffer to the reaction solution after 30 min (terminal point of the reaction). After the reaction, the droplet was separated from the monolith and brought onto the hydrophilic SU-8 spot. Next, the top plate was removed and the reaction solution was let dry prior to DAPPI-MS analysis. The specificity of the DAPPI-MS identification of phenacetin metabolites was confirmed with help of blank incubation matrix, i.e., control reactions performed in the absence of phenacetin.

## 3. Results and Discussion

### 3.1. Material Considerations

The feasibility of SU-8 for low-cost chip fabrication by spin-coating and photolithographic patterning has raised considerable interest in its use as the dielectric material in DMF devices. SU-8 not only provides mechanically strong films with high thermal and chemical stability, but it also has notably high dielectric strength (4.4 MV/cm [31]), similar to that of parylene C (the most used dielectric in DMF). In this work, the DMF driving voltages typically ranged between 80 and 120 V_rms_ at 10 kHz (for actuation of aqueous solutions). Under these conditions, each chip could be used for several subsequent droplet manipulations and for long term (here, 30 min) continuous actuation, such as mixing of the drug distribution assays.

In addition to high dielectric strength, the good uniformity and the high crosslinking degree of the cured SU-8 layer (film) are also in a critical role in terms of the robustness of SU-8 based DMF devices. Namely, when compared with another commercial negative tone photoresist, OrmoComp^®^, showing equally high dielectric strength [32], we observed that, while SU-8 based devices remained undamaged, the OrmoComp^®^ based devices suffered from frequent dielectric breakdown between the electrodes. As the result of the electric discharge, discoloration of the dielectric OrmoComp layer was often visible even by eye as illustrated in Appendix A (see brownish color at the edges of electrodes). The dielectric breakdown typically occurred in the proximity of the edges of the metal patterns, e.g., between the active electrode and its adjacent electrode (grounded), which were the most challenging spots of our assay design in terms of the layer quality. Even if both materials were applied (by spin-coating) and cured in the same way, the good processability of SU-8 was shown to result in superior performance over OrmoComp in electrowetting. Furthermore, SU-8 tolerated well the harsh desorption/ionization conditions characteristic to DAPPI-MS. Namely, in DAPPI-MS, sample desorption is induced by a heated (>200 °C) jet of vaporized organic solvent (Figure 3a), which sets strict requirements for the stability of the substrate material. However, the heated solvent jet did not cause any visible defects or degradation of the SU-8 surface within the normal desorption time (1–2 min) as evidenced by the lack of interfering background ions in the MS spectra (Figure 3b). Although the fluoropolymer surface gave equally low background interference (see Appendix A) as that of SU-8, the sample precipitation on the fluoropolymer posed substantial challenges in terms of repeatability as explicated below.

One of the critical factors with a view to successful ambient MS analysis on artificial (not tissue or plant derived) surfaces is the uniformity of the sample precipitation pattern. Besides the compound’s solubility (to the evaporated solvent), the surface’s wetting properties play a key role in facilitating controlled precipitation of the sample solution [33]. In case of aqueous sample solutions, the high water contact angle of the fluoropolymer coating (top most layer of the DMF devices) results in a somewhat dense precipitation pattern because of decreasing droplet size upon evaporation (Figure 3c). This sets limitations on the feasibility of the hydrophobic coating for desorption/ionization, because very small precipitate complicates the alignment of the heated solvent jet to the sample spot and thus results in reduced repeatability. Instead by exposing hydrophilic SU-8 spots into the DMF chip design, we were able to control the droplet size upon evaporation with help of surface tension, which resulted in a uniform and less dense precipitation pattern regardless of the solvent composition (Figure 3c). In this study, we mainly compared the sample precipitation from fully aqueous solvent (needed for drug distribution and metabolism assays) and from water-methanol 1:1 solution (used for method validation). No significant difference was observed in the precipitation patterns between these two solvents. Instead, desorption/ionization efficiency and repeatability were shown to be much better on top of SU-8 (Figure 3d) than on the fluoropolymer (Figure 3e). This was likewise because of the homogenous precipitate and the fact that the edges of the SU-8 spots facilitated more precise visual alignment of the heated vapor jet to the sample spot. The hydrophilic spots also allowed us to ‘anchor’ the droplets to the bottom plate, when removing the top plate of the DMF device to allow for evaporation prior to ambient MS analysis. On the basis of these qualitative comparisons, SU-8 was concluded well feasible for fabrication of DMF microdevices even under non-cleanroom conditions as well as for multiple repeated use (of the hydrophilic SU-8 spots) in DAPPI-MS.

### 3.2. Mass Spectrometry Method Qualification

The lower limits of detection (LLODs) of the three model compounds selected were determined with help of a series of calibration samples ranging between 0.1 and 10 µg/mL (each compound, in methanol-water 1:1) and were 1 µg/mL (7 pmol) for paracetamol, 0.25 µg/mL (1 pmol) for naproxen, and 0.5 µg/mL (2 pmol) for carbamazepine. Of these, paracetamol and carbamazepine (as well as the internal qualifier, testosterone) were detected as protonated [M + H]^+^ ions and naproxen as a radical M^+•^ cation. The LLODs were calculated based on five replicates spots at each concentration. LLOD was defined as the lowest concentration that gave detectable signal (integrated peak) from all five spots. As such, the method was concluded feasible for, e.g., detection of drugs in bodily fluids, such as plasma or urine, which typically feature drug concentrations in the range of few µg/mL. However, it should be noted that DAPPI-MS is inherently not a quantitative analysis method, but best applicable for qualitative monitoring of in vivo administered drugs or identification of the in vitro reaction products (as will be shown in the next chapters). Minor variation in the ion current intensity between parallel spots as in Figure 3e is also inherent to DAPPI-MS. For comparison, the repeatability of paracetamol (66 pmol) signal intensity on SU-8 and on polymethylmetacrylate (a common standard substrate for DAPPI-MS [26]) were 35% (RSD, n = 5 spots) and 23% (RSD, n = 5 spots), respectively.

To evaluate the reuse possibility of the DMF chips, i.e., the cross-contamination risk between subsequent samples applied onto the same hydrophilic SU-8 spot, we also compared the background spectra and ion intensities between fresh (unused) and cleaned (reused) hydrophilic spots. Cleaning of the spots between experiments was carried out by rinsing the spots manually in methanol and water, which is our standard procedure for cleaning of the DMF chips between subsequent uses. As illustrated in Appendix A, there were no traces of standards on the spots after washing and the intensities of the recorded signals were within the normal run-to-run variation before and after washing (Appendix A).

### 3.3. On-Chip DAPPI-MS Analysis of Drug Distribution Assays

Octanol-water distribution, i.e., logD coefficient, is a routine tool for assessing the solubility of a drug compound. Solubility is not only a critical factor to distinguish between continuation and termination of the development of a new drug candidate, but it also serves as the basis for sample purification possibilities by liquid-liquid extraction (LLE). In this work, we demonstrated the feasibility of the DMF-DAPPI-MS combination for preliminary examination of a drug’s solubility and distribution properties by conducting a three-phase (aqueous → organic → aqueous) LLE on chip and monitoring the distribution of three model compounds (naproxen, paracetamol, and carbamazepine) between the organic solvent and aqueous phases. The LLE was performed in two steps to extract the selected pharmaceuticals first from aqueous (acidic) to organic (octanol) solution and then from the organic solvent back to an aqueous (alkaline) solution. Since octanol can be actuated only at lower frequencies (300 Hz) in DMF device, it was possible to selectively actuate the aqueous droplets only by using higher frequency (10 kHz) to bring them inside and outside of the larger, immobile octanol droplet (Appendix A). By encapsulating the aqueous droplets inside octanol, the assay was also insensitive to evaporation effects owing to the very low evaporation rate of octanol.

All selected test compounds possessed acidic nature (pK_a_ ranging between 4.19 and 15.96, Table 1) and were thus in neutral form in the first aqueous donor solution (0.1% TFA, pH 2), which facilitated their extraction to the organic acceptor solution (octanol). However, owing to their varying water solubility and acidity (Table 1), the compounds’ distribution from the organic phase and back to the alkaline aqueous phase varied according to these parameters. Naproxen was the most acidic compound (pKa 4.19) with moderate lipophilicity (logD 2.98) at pH 2 and low lipophilicity (logD −0.54) at pH 12, and thus, it was efficiently extracted from aqueous to organic phase (in a neutral form) and further from organic to aqueous phase (in an ionized form). Paracetamol on the other hand was more water soluble (logD 0.91) at pH 2, which reduced its extraction efficiency from the first aqueous donor solution to the octanol phase compared with naproxen as illustrated in Figure 4a. Nevertheless, abundant naproxen and paracetamol peaks were detected by DAPPI-MS after evaporation of the final acceptor solution (Figure 4b). Instead carbamazepine as the least water-soluble (logD 2.76) and only weakly acidic (pKa 15.96) compound was not back-extracted from octanol to the alkaline aqueous phase and was thus not detected at all in the final acceptor solution. Applying mixing (by DMF actuation of the octanol phase at 300 Hz) during LLE did not much affect the extraction yields in any of the cases (data not shown), because the distribution processes only occur at the water-octanol interface.

### 3.4. On-Chip DAPPI-MS Analysis of Drug Metabolism Assays

Drug metabolism assays are typically conducted at a physiological pH (7–7.5) in the presence of a relatively high salt concentration to preserve the enzymatic activity. The most commonly used electrolytes in CYP enzyme assays include phosphate and Tris buffers. However, phosphate is inherently not feasible for MS analysis, because of its nonvolatile nature. Upon sample evaporation, the use of a phosphate buffer also results in a very dense precipitate, which would impair ambient MS analysis independent of the chosen desorption/ionization method. Therefore, we chose to conduct the on-chip CYP assays by using the more volatile Tris (base) as the buffer component and determined its impact on the ionization of paracetamol (the model metabolite produced via CYP1A2 induced phenacetin O-dealkylation). Although some variation between the average (n = 5 samples) paracetamol intensities were observed between pure solvent (methanol-water 1:1) and the Tris buffer, these were within the normal spot-to-spot variation of DAPPI-MS (Figure 4e). However, in Tris buffer, the protonated paracetamol ion [M + H]^+^ = 152.071 was dominated by the protonated Tris adduct ion [M + Tris + H]^+^ = 273.199. Nevertheless, the sensitivity of DAPPI-MS analysis directly from the DMF bottom plate was shown to be good enough to facilitate detection of the trace-level amounts (few pmol) of drug metabolites produced in the droplet based CYP reactions in situ, as illustrated in Figure 4f. As seen in Figure 4f, no metabolite was observed in blank control assay (with no substrate) whereas paracetamol Tris adduct ion peak was detected in all positive control assays.

## 4. Conclusions

MS analysis of droplet-scale (~µL) biochemical reactions is often compromised by the sample loss associated with off-chip handling of small volumes, which may be required for, e.g., purification of sample from biological buffers or transfer of the sample onto a conventional LC-MS separation system. Ambient MS is therefore an appealing alternative to facile and rapid detection of droplet-based assays directly on the surfaces of DMF devices. Besides the fact that excessive sample purification can be omitted, the combination also benefits from the small sample volume, which allows rapid evaporation of the sample by simply removing the top plate of the DMF device prior to desorption/ionization.

This work describes the feasibility of DAPPI-MS for instant analysis of sample components directly from the surface of a DMF based sample preparation device. With help of DAPPI-MS, the sample components can be identified in situ without need for integration of additional detector elements (e.g., ESI emitters) or transfer of the sample droplets for off-chip analysis. The successful combination of DMF and DAPPI-MS was promoted by the favorable mechanical, thermal, and chemical properties of SU-8, which served a bifunctional role, as dielectric material for electrowetting and as substrate material for desorption/ionization. Owing to their high dielectric and mechanical strength, SU-8 dielectric layers showed robust performance in DMF even upon prolonged continuous use. On the other hand, the hydrophilic nature of SU-8 (spots exposed to the assay design) allowed us to control the size and uniformity of the sample precipitate upon evaporation of sample droplet on top of the SU-8 spot. As the result, we were able to detect as low as pmol amounts of drugs per droplet (corresponding to less than µg/mL sample concentrations) by the DMF-DAPPI-MS setup, suggesting that the method is readily feasible for detection of, e.g., in vivo administered drug concentrations in bodily fluids. Most importantly, the good chemical and thermal stability of SU-8 facilitated multiple desorption/ionization cycles (repeated use) of each SU-8 spot without any visible damage of the substrate surface. In all, the straightforward integration of DMF with DAPPI-MS increases the flexibility of the analysis of DMF assays by facilitating identification of the sample components in situ directly on the DMF bottom plate. Moreover, the chip fabrication protocol developed herein can be carried out under standard laboratory conditions, which significantly reduces the fabrication costs and makes rapid prototyping of the DMF chips much easier compared with conventional parylene C based devices.

In all, these technical advances may pave the way for adaptation of the droplet technology in preclinical drug discovery and development. Even if quantitation of the produced drug metabolites or drug distribution ratios is not feasible because of the qualitative nature of the DAPPI-MS analysis, the possibility to categorize new drug candidates according to their solubility using the droplet scale liquid-liquid extraction experiments could allow for a cost-efficient tool to distinguish between poorly and highly soluble drugs (a key parameter for continuation vs. termination of the development of new drug candidates). Alternatively, at a later stage of drug development, preliminary identification of the most abundant drug metabolites isoenzyme-specifically could allow for more efficient design of experiments and thus result in substantial savings in the analysis costs. Owing to the inherent possibility of DMF for parallelism of multiple individual reactions (each with different CYP isoenzyme), the metabolic profile of a new drug candidate could be preliminarily examined with help of the droplet reactions at much lower cost and reagent consumption compared with the conventional liquid chromatography mass spectrometry protocols. Thereby, the developed DMF-DAPPI-MS setup could allow for competitive advantage to rapid, preliminary characterization of new drug candidates with a view to their metabolism and disposition before more in-depth studies.

## Figures and Tables

**Figure 1 micromachines-09-00649-f001:**
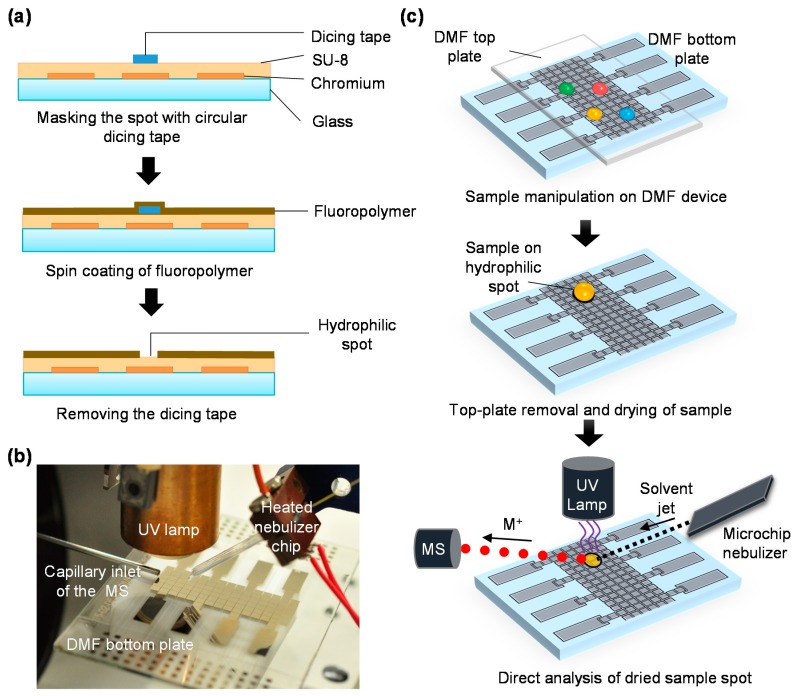
(**a**) Schematic view of the fabrication process of digital microfluidic (DMF) bottom plate incorporating hydrophilic spots, (**b**) a photograph of DMF-desorption atmospheric pressure photoionization (DAPPI) interface, and (**c**) schematic representation of the process flow of DMF-DAPPI-mass spectrometry (MS) analysis. For details of the heated nebulizer chip fabrication and use, see references [24,28], respectively.

**Figure 2 micromachines-09-00649-f002:**
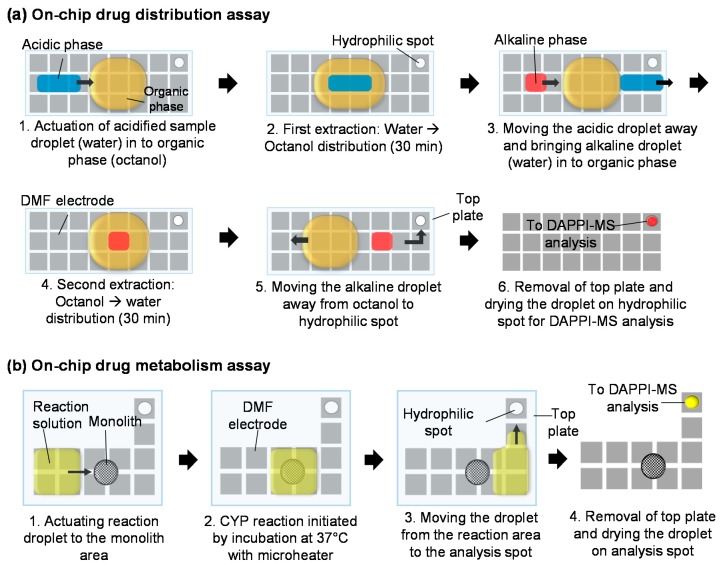
Schematic views of the main steps of (**a**) the on-chip drug distribution assays and (**b**) the on-chip drug metabolism assays.

**Figure 3 micromachines-09-00649-f003:**
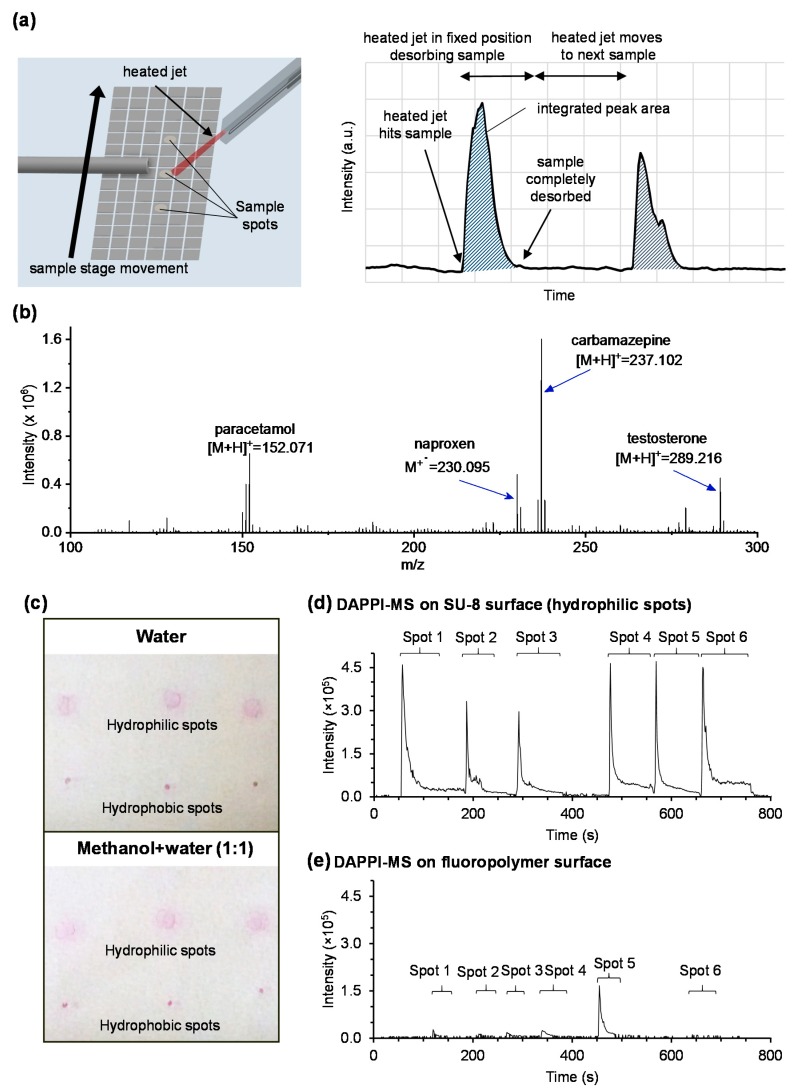
(**a**) Schematic illustration of the process of (**a**) desorption/ionization of samples from DMF chip surface, and integration of peak area of extracted ion chromatogram (EIC) in DAPPI-MS analysis. (**b**) Mass spectrum of paracetamol (66 pmol), naproxen (44 pmol), carbamazepine (42 pmol), and testosterone (70 pmol) dissolved in methanol-water 1:1, deposited onto the hydrophilic SU-8 spot, and analyzed by DAPPI-MS. (**c**) Photograph of rhodamine B precipitation pattern on hydrophilic SU-8 spots and on hydrophobic fluoropolymer in two different solvents: water (top) and methanol-water 1:1 (bottom). (**d**,**e**) EICs of testosterone (internal qualifier, 70 pmol in methanol-water 1:1) deposited and analyzed from parallel spots (n = 6) by DAPPI-MS: (**d**) on hydrophilic SU-8 spots, and (**e**) on hydrophobic fluoropolymer.

**Figure 4 micromachines-09-00649-f004:**
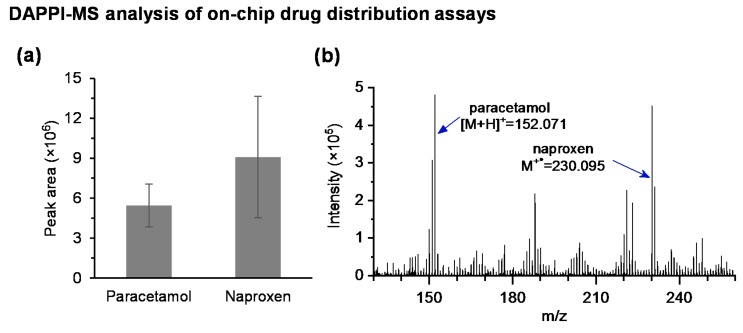
(**a**) Comparison of the repeatability (n = 3 LLE experiments) of intensities of extracted paracetamol (initial amount 300 pmol) and naproxen (initial amount 200 pmol) after LLE and evaporation of the final aqueous acceptor solution. (**b**) Mass spectra of the final acceptor solution featuring abundant paracetamol and naproxen ions. (**c**) Illustration of monolith functionalization and phenacetin-O-deethylation reaction and (**d**) scanning electron microscope image of porous thiol-ene monolith. (**e**) Comparison of paracetamol signal intensities in pure solvent (methanol-water 1:1) vs. 60 mM Tris buffer (n = 5 samples each). (**f**) EICs of paracetamol (red line) produced via CYP1A2 mediated phenacetin (black line) O-dealkylation analyzed from the dried precipitate of the enzymatic reaction solution after the on-chip drug metabolism assays.

**Table 1 micromachines-09-00649-t001:** The chemical structures, monoisotopic masses and chemical properties of the model pharmaceuticals used. Molecular properties from Chemicalize (https://chemicalize.com/) developed by ChemAxon Ltd. (Budapest, Hungary).

**Compound**	Paracetamol 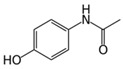	Naproxen 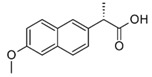	Carbamazepine 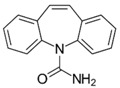	Testosterone 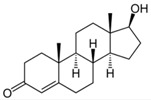
**Molecular Formula**	C_8_H_9_NO_2_	C_14_H_14_O_3_	C_15_H_12_N_2_O	C_19_H_28_O_2_
**Observed Ion Type**	152.071 ([M + H]^+^)	230.095 (M^+•^)	237.102 ([M + H]^+^)	289.216 ([M + H]^+^)
**Monoisotopic Mass (g/mol)**	151.063	230.094	236.095	288.209
***p*K_a_ (acidic)**	9.46	4.19	15.96	n.d. (internal qualifier)
**LogD (pH 2)** **LogD (pH 12)**	0.9 −1.15	2.98 −0.54	2.76 2.76	

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
