# Peer review of "Interfacing Digital Microfluidics with Ambient Mass Spectrometry Using SU-8 as Dielectric Layer"

_micromachines, 2018, doi:10.3390/mi9120649_

Round 1

Reviewer 1 Report

The authors demonstrated a digital microfluidic (DMF) chip design that is compatible with desorption atmospheric pressure photoionization mass spectrometry (DAPPI-MS). The fabrication can be performed without a cleanroom, using solution-processable SU-8 dielectric and tape-patterned fluoropolymer coating. The authors characterized the electric and thermal stability of SU-8 for droplet actuation and DAPPI-MS analysis, respectively. Further, they demonstrated the importance of hydrophilic patterns for the reproducibility of the analysis. As a proof of concept, the authors performed two types of on-chip assays: 1) drug distribution and 2) metabolism. These assays highlight the versatility of digital microfluidics for sample preparation, integrating liquid-liquid extraction or solid-phase/heated reaction on the front end of DAPPI-MS. Overall, the manuscript is thorough and the new concepts will be of great interest to the field.

Minor points to address:

1)      The authors appeared to have compared water vs. methanol-water (1:1) for sample drying in the DAPPI-MS analysis (Figure 3c), but have not discussed why methanol-water was chosen in subsequent experiments.

2)      The assay performed here only results in a qualitative binary output. Ideally, the distribution assay should enable LogP/LogD determination that is comparable to macroscale-determined values. The metabolic assay should enable one to learn something about the enzyme or determine novel metabolites of a drug. Overall, the authors succeeded in demonstrating that different types of DMF sample-processing can be integrated with DAPPI-MS, but it is unclear what the author ultimately wants to achieve with this technology. Thus, it would be useful for the authors to discuss the problems/unmet need this technology will address in the future, and what are the remaining challenges to get there.

3)      This is similar to #2. The authors reported some lower limits of detection for each drug. Given this performance, what types of applications is this useful for?

Author Response

Point 1: The authors appeared to have compared water vs. methanol-water (1:1) for sample drying in the DAPPI-MS analysis (Figure 3c), but have not discussed why methanol-water was chosen in subsequent experiments.

Response 1: The comparison of the precipitation patterns with water vs. water-methanol were included Figure 3c, because both of these solvents were used in the experiments. The method validation was conducted by using drug standards dissolved in methanol-water, whereas the drug distribution and metabolism assays were carried out in fully aqueous media. As illustrated in Figure 3C, no significant difference was observed in the evaporation/precipitation patterns between these two solvents. This discussion has now been added to the manuscript (page 9, lines 283-286).

Point 2: The assay performed here only results in a qualitative binary output. Ideally, the distribution assay should enable LogP/LogD determination that is comparable to macroscale-determined values. The metabolic assay should enable one to learn something about the enzyme or determine novel metabolites of a drug. Overall, the authors succeeded in demonstrating that different types of DMF sample-processing can be integrated with DAPPI-MS, but it is unclear what the author ultimately wants to achieve with this technology. Thus, it would be useful for the authors to discuss the problems/unmet need this technology will address in the future, and what are the remaining challenges to get there.

Response 2: We thank the Reviewer for insightful comments and have revised the discussion (page 9, lines 304-308 and page10, line 323) and the conclusions (page 12, line 403 and page 13, lines 412-427) about the envisioned needs in the field of drug discovery and development that could perhaps be met with the developed technology. Briefly, the developed DMF-DAPPI setup is foreseen to allow for competitive advantage to rapid, preliminary characterization of new drug candidates with a view to their metabolism and disposition before in-depth liquid chromatography mass spectrometry (LC-MS) studies. Even if quantitation of the produced drug metabolites or drug distribution ratios is not feasible because of the qualitative (binary) nature of the DAPPI-MS analysis, the possibility to categorize new drug candidates according to their solubility using the droplet scale liquid-liquid extraction experiments could allow for a cost-efficient tool to distinguish between poorly and highly soluble drugs, which is often the key parameter for continuation vs. termination of the development of new drug candidates. In other words, the development of compounds with low solubility is often not continued because of the problems associated with their poor distribution properties affecting the reliability of in vitro and in vivo assays. Similarly, at a later stage of drug development, preliminary identification of the most abundant drug metabolites isoenzyme-specifically could allow for more efficient design of LC-MS experiments and thus result in substantial savings in the analysis costs. Owing to the inherent possibility of DMF for parallelism of multiple individual reactions (each with different CYP isoenzyme), the metabolic profile of a new drug candidate could be preliminarily examined with help of the droplet reactions at much lower cost and reagent consumption compared with the conventional enzyme incubation and LC-MS protocols. In all, it is the low cost and the rapid response achieved with the DMF-DAPPI-MS technology in preliminary drug testing, which provides competitive edge over conventional analytical protocols.

Point 3 :This is similar to #2. The authors reported some lower limits of detection for each drug. Given this performance, what types of applications is this useful for?

Response 3: Given the fact that as low as picomole amounts of drugs could be detected on the DMF surface by DAPPI-MS, the technology is readily feasible for, e.g., analysis of drug compounds in bodily fluids (typically few mg/mL) in addition to the in vitro applications described in the context of the previous response to Q2. These applications are now briefly discussed in the “Conclusions” section.

Reviewer 2 Report

This is a very interesting study that uses a Digital Microfluidic platform (the DropBot system from the Wheeler’s lab) for DAPPI-MS. DAPPI-MS is exciting, being integrated with the droplet automation technology to this sample analysis tool will further facilitate the automation of the conventional laboratory operations.

The concern was if the heated solvent will destroy the SU-8 and Fluoropolymer coating or introduce background to the MS reading. The authors mentioned at the bottom of Page 14 that the MS spectra did not indicate any SU-8 background. However, what about Fluoropolymer?

Can you explain what are the brown-color spots in Figure S2?

Does this system have problems in the alignment of the Capillary inlet of the MS, the chip, the Heated nebulizer, and the UV lamp? Would the misalignment make the results unrepeatable?

The 8-um SU-8 may require higher driving voltage than 80 – 120 V for a smooth droplet actuation. Did the authors try higher voltages? Will that be helpful for a better droplet movement? What is the lifespan of the hydrophobic coating using different driving voltages?

An extra short video or a series of pictures of the droplet actuation will be very helpful for the readers to understand the device and the operation.

Author Response

Point 1: The concern was if the heated solvent will destroy the SU-8 and Fluoropolymer coating or introduce background to the MS reading. The authors mentioned at the bottom of Page 14 that the MS spectra did not indicate any SU-8 background. However, what about Fluoropolymer?

Response 1: The background spectra of both fluoropolymer and SU-8 surfaces are now added to the supplementary material as Figure S3, which is also cited in the main manuscript on page 7, line 259. The background spectrum of fluoropolymer was somewhat identical to that of SU-8, but the high hydrophobicity of the fluoropolymer layer resulted in too dense sample precipitate, which impaired the efficiency of desorption/ionization on fluoropolymer. This discussion has been added to the manuscript on page 7 lines 259-261.

Point 2: Can you explain what are the brown-color spots in Figure S2?

Response 2: As mentioned in the manuscript (page 7, lines 244-251) and in the Supplementary material (page 1, under “Characterization of the SU-8 and OrmoComp dielectric layers”), the brownish discoloration is due to the electric discharge in the dielectric (OrmoComp) layer between two adjacent electrodes. This explanation has been clarified to the manuscript (page 7, line 248).

Point 3: Does this system have problems in the alignment of the Capillary inlet of the MS, the chip, the Heated nebulizer, and the UV lamp? Would the misalignment make the results unrepeatable?

Response 3: The capillary inlet of the MS and the UV lamp are aligned and kept at fixed positions, whereas the DMF chip and the heated jet produced by the nebulizer chip are typically aligned just prior to analysis with help of an xyz alignment tool. Obviously, the (mis)alignment has a huge impact on the ionization efficiency and repeatability. Poorer alignment possibilities were indeed one of the main reasons why the repeatability of analysis was much poorer on fluoropolymer (tiny, dense precipitate) than on SU-8 (larger, homogenous precipitate), as already explained in the manuscript on page 9, lines 275-283. However, in case the sample was evaporated on SU-8, the edges of the hydrophilic patterns themselves facilitated improved alignment accuracy over non-patterned surfaces. This has now been clarified in the manuscript (page 9 lines 288-290).

Point 4: The 8-um SU-8 may require higher driving voltage than 80 – 120 V for a smooth droplet actuation. Did the authors try higher voltages? Will that be helpful for a better droplet movement? What is the lifespan of the hydrophobic coating using different driving voltages?

Response 4: On an average, to actuate an aqueous droplet on 8-µm-thick-SU-8 coated DMF chip in 2-plate configuration, 100-120 Vrms is sufficient. Droplet movement with similar voltages on 8-µm-thick-SU-8 was achieved in our previous work (Sathyanarayanan et al., Anal. Bioanal. Chem., 2018, manuscript reference 7) and also by others (Dryden et al., Anal. Chem., 2013, manuscript reference 10). It is commonly known, that under these voltages, the fluoropolymer does not suffer from any damage. Instead, increasing the driving voltage will typically shorten the lifespan of the dielectric (or the fluoropolymer) coating and therefore we did not exceed the above mentioned voltages even if theoretically the droplet should move a bit faster at higher voltages. We have not performed a systematic test in terms of the device lifespan vs. driving voltages, but a qualitative estimate of the durability of the coatings is given on page 7, lines 239-241. No changes were made to the manuscript regarding this question.

Point 5: An extra short video or a series of pictures of the droplet actuation will be very helpful for the readers to understand the device and the operation.

Response 5: As suggested, a short video of droplet actuation is added as supplementary video S1 and cited in page 10, line 334. The video depicts the actuation of an aqueous droplet into an octanol droplet in drug distribution (liquid-liquid extraction) assays.

Round 2

Reviewer 1 Report

no further comments